# Systemic Cytokines in Retinopathy of Prematurity

**DOI:** 10.3390/jpm13020291

**Published:** 2023-02-05

**Authors:** Po-Yi Wu, Yuan-Kai Fu, Rey-In Lien, Ming-Chou Chiang, Chien-Chung Lee, Hung-Chi Chen, Yi-Jen Hsueh, Kuan-Jen Chen, Nan-Kai Wang, Laura Liu, Yen-Po Chen, Yih-Shiou Hwang, Chi-Chun Lai, Wei-Chi Wu

**Affiliations:** 1Department of Education, Chang Gung Memorial Hospital, Linkou Branch, Taoyuan 333, Taiwan; 2School of Medicine, National Yang Ming Chiao Tung University, Taipei 112, Taiwan; 3Division of Neonatology, Department of Pediatrics, Chang Gung Memorial Hospital, Linkou Branch, Taoyuan 333, Taiwan; 4College of Medicine, Chang Gung University, Taoyuan 333, Taiwan; 5Department of Ophthalmology, Chang Gung Memorial Hospital, Linkou Branch, Taoyuan 333, Taiwan; 6Center for Tissue Engineering, Chang Gung Memorial Hospital, Linkou Branch, Taoyuan 333, Taiwan; 7Department of Ophthalmology, Edward S. Harkness Eye Institute, Columbia University Irving Medical Center, Columbia University, 622 W 168th St, New York, NY 10032, USA; 8Department of Ophthalmology, Tucheng Municipal Hospital, New Taipei 236, Taiwan; 9Department of Ophthalmology, Jen-Ai Hospital Dali Branch, Taichung 412, Taiwan; 10Department of Ophthalmology, Chang Gung Memorial Hospital, Keelung Branch, Keelung 204, Taiwan

**Keywords:** cytokine, inflammation, neovascularization, oxygen-induced retinopathy, preterm infant, retinopathy of prematurity

## Abstract

Retinopathy of prematurity (ROP), a vasoproliferative vitreoretinal disorder, is the leading cause of childhood blindness worldwide. Although angiogenic pathways have been the main focus, cytokine-mediated inflammation is also involved in ROP etiology. Herein, we illustrate the characteristics and actions of all cytokines involved in ROP pathogenesis. The two-phase (vaso-obliteration followed by vasoproliferation) theory outlines the evaluation of cytokines in a time-dependent manner. Levels of cytokines may even differ between the blood and the vitreous. Data from animal models of oxygen-induced retinopathy are also valuable. Although conventional cryotherapy and laser photocoagulation are well established and anti-vascular endothelial growth factor agents are available, less destructive novel therapeutics that can precisely target the signaling pathways are required. Linking the cytokines involved in ROP to other maternal and neonatal diseases and conditions provides insights into the management of ROP. Suppressing disordered retinal angiogenesis via the modulation of hypoxia-inducible factor, supplementation of insulin-like growth factor (IGF)-1/IGF-binding protein 3 complex, erythropoietin, and its derivatives, polyunsaturated fatty acids, and inhibition of secretogranin III have attracted the attention of researchers. Recently, gut microbiota modulation, non-coding RNAs, and gene therapies have shown promise in regulating ROP. These emerging therapeutics can be used to treat preterm infants with ROP.

## 1. Introduction

Retinopathy of prematurity (ROP) is a developmental proliferative vascular disorder of the retina that affects 30–50% of preterm infants with very low birth weight (BW) [1]. Low BW, gestational age (GA), and postnatal exposure to unregulated high or fluctuating oxygen levels are the most commonly recognized risk factors for ROP [1]. With the greater survival of premature infants due to improved perinatal care, ROP has increasingly become the leading cause of preventable childhood blindness worldwide [1].

Human retinal vascularization commences from the optic disc around the 16th week of gestation, extends centrifugally to the peripheral retina, and finalizes at the 40th week of gestation [2]. The process is interrupted by preterm birth, leaving a peripheral avascular zone whose size depends on the immaturity of the neonate [2]. The pathophysiology of ROP has been studied not only in humans but also in many animal models of oxygen-induced retinopathy (OIR) to identify potential treatment modalities [3]. ROP development can be divided into two phases: vaso-obliteration (phase 1) and vasoproliferation (phase 2) [3]. During phase 1, retinal microvascular degeneration occurs because of hyperoxia-induced damage, resulting in the peripheral avascular retina [3]. As hypoperfused tissue cannot meet the increasing metabolic demands of the developing retina, phase 2 of the disease is initiated. Under hypoxia, abnormal neovascularization is triggered by the release of growth factors, leading to the misdirection of vessel growth from the retina into the vitreous [3]. Consequently, fibrovascular traction may cause partial or total retinal detachment, subsequently causing severe visual impairment [3].

Conventional treatments include ablative cryotherapy and laser photocoagulation, which directly damage the avascular retina, thereby minimizing the secretion of the growth factors responsible for neovascularization [4]. However, neither treatment specifically targets the molecular pathways of pathological neovascularization. Although both treatments decrease the risk of blindness, they result in numerous unfavorable sequelae, such as inflammation, myopia, macular dragging, loss of peripheral visual field, and scar induction [4,5,6,7].

Recently, intravitreal anti-vascular endothelial growth factor (VEGF) administration has shown promising results in resolving ROP, particularly in zone 1 disease [8]. However, various issues, including neovascularization recurrence as late as 60 weeks of postmenstrual age (PMA) [9], long-term deficits in photoreceptor integrity [10], and potential systemic toxicity hampering organogenesis [11,12,13,14], remain a great concern for ROP treatment. Some patients are refractory to anti-VEGF treatment [15], implying the importance of identifying other angiogenic or anti-angiogenic cytokines involved in the pathogenesis of ROP.

Inflammation resulting from various perinatal and neonatal insults has recently been implicated in the pathogenesis of ROP [1]. The actions of systemic cytokines, chemokines, growth factors, and immune cells, such as leukocytes, monocytes, and macrophages/microglia, may interfere with retinal vasculature development in time- and dose-dependent manners [16,17]. The present review aims to illustrate the characteristics and actions of all cytokines participating in the pathogenesis of ROP as well as the underlying signaling pathways and cellular responses, associated perinatal insults, and novel pharmacological agents to target these cytokine signaling pathways.

## 2. Cytokines: Characteristics and Actions

### 2.1. Interleukin (IL)-1β

IL-1β, a crucial mediator of the inflammatory response, is known for its involvement in the development of vasoproliferative retinopathies [18]. Under ischemic conditions, IL-1β levels are significantly elevated in recruited neutrophils, endothelial cells, and retinal glial cells [19]. In the early stages of rat OIR, microglia-derived IL-1β sustains the activation of microglia and induces microvascular injury by the release of semaphorin-3A from adjacent neurons [18]. In rat models, IL-1β, along with the tumor necrosis factor-alpha (TNF-α), also triggers retinal ganglion cell death and breakdown of the blood–retina barrier (BRB) [20,21]. Inhibition of the IL-1β receptor leads to a marked decrease in vaso-obliteration and subsequent pathological neovascularization in OIR [18]. IL-1β receptor antagonists have been reported to prevent choroidal involution and other long-term outer neuroretinal anomalies in another rat model of OIR [22]. However, vitreous IL-1β levels have been reported to be comparable and below detectable levels in preterm infants with ROP and their non-ROP counterparts, respectively [23].

### 2.2. IL-6

IL-6 is involved in infections and tissue injuries via the stimulation of the acute-phase protein response, hematopoiesis, and immune reactions [24]. It has also been shown to promote VEGF expression [25]. It possesses not only pro-inflammatory but also anti-inflammatory properties [26]. Elevated IL-6 levels have been observed in both the maternal plasma and plasma/serum of preterm infants around birth, who later developed mild or severe ROP [27,28,29,30]. The cord plasma IL-6 level has been reported to be an independent marker for predicting severe ROP [31].

### 2.3. IL-7

IL-7 participates in host defense by regulating the development and homeostasis of T, B, and natural killer cells [32]. In vitro, IL-7 is capable of inducing RPE-derived monocyte chemotactic protein-1 (MCP-1) and IL-8 [33]. Its pro-inflammatory properties have been implicated in the pathogenesis of neovascular age-related macular degeneration (AMD) and diabetic retinopathy [34,35]. One study reported the expression of IL-7 in cord blood associated with ROP development in preterm infants [36], and another study reported significantly high vitreous IL-7 levels in the ROP eyes at the time of vitrectomy [23].

### 2.4. IL-8

IL-8, the first investigated chemokine, has important effects on angiogenic activity and the induction of ocular inflammation [37]. Its unique pro-angiogenic properties include (1) chemotaxis via directional migration of neutrophils, basophils, and T lymphocytes, (2) stimulation of both endothelial proliferation and capillary tube formation, (3) inhibition of endothelial cell apoptosis, and (4) enhancement of matrix metalloproteinase (MMP)- 2, MMP-9, and gelatinase activity [37]. In a rat OIR model, increased expression of an IL-8 homologue was noted during the peak time points of neovascularization [38]. Accordingly, a few studies have also demonstrated the correlation between high plasma/serum IL-8 levels in preterm infants after birth and the later development of severe ROP [30,39,40].

### 2.5. IL-10

IL-10 can suppress the inflammatory response induced in microglial cells by inhibiting the release of TNF-α, MIP-1α, and Regulated upon Activation, Normal T cell Expressed and Secreted (RANTES) in vitro [41]. In an OIR mouse model, however, IL-10 was found to promote pathological angiogenesis by guiding macrophage behavior toward a pro-angiogenic phenotype [42].

### 2.6. IL-17

IL-17 is a pro-inflammatory cytokine that primarily provides protection against extracellular bacterial and fungal pathogens [43]. It also plays a crucial role in intraocular inflammation [44]. In a mouse OIR model, blocking IL-17 decreased the retinal areas of nonperfusion and neovascularization [45]. However, low serum IL-17 levels at birth have been noted in preterm infants who later developed severe ROP [27].

### 2.7. IL-18

IL-18, a pro-inflammatory cytokine, acts as an immunoregulator with both angiogenic and angiostatic properties [46]. In a mouse OIR model, IL-18 was found to regress retinal pathological neovascularization rather than inhibit its development [46]. Serum IL-18 levels have been reported to be lower in preterm infants with ROP than in those without ROP, but the levels increased within three weeks of birth [27]. This biphasic pattern of IL-18 highlights its role as a modulator of angiogenesis with time-sensitive expression in different phases of ROP [27].

### 2.8. IL-19

IL-19, a proposed anti-inflammatory cytokine, has been hypothesized to serve as a compensatory mediator in response to inflammatory stimuli [47]. However, a recent study found that IL-19 was able to enhance pathological neovascularization by promoting pro-angiogenic M2 macrophage polarization in mouse OIR [48]. In vitro, IL-19 also induces the proliferation and migration of human retinal endothelial cells [48].

### 2.9. IL-1 Receptor Antagonist (IL-1Ra)

IL-1Ra levels are significantly increased in the vitreous humor and tears of preterm infants with ROP, together with elevated levels of VEGF, complement component proteins, and MMP-9 [49]. Wooff et al. [50] hypothesized that increased IL-1Ra expression is a compensatory mechanism against the angiogenic effects of IL-18 and the deleterious effects of IL-1β.

### 2.10. TNF-α

Predominantly originating from monocytes or macrophages, TNF-α is the primary initiator of inflammation [51]. It enhances the production of other cytokines, such as IL-8, basic fibroblast growth factor (bFGF), and MCP-1, in retinal microglia adjacent to vessels in an autocrine or paracrine manner [52]. Its aforementioned functions, including triggering ganglion cell death and the breakdown of BRB, have been implicated in the hypoxic retina [20,21]. In mouse OIR models, inhibition of TNF-α markedly improves vascular recovery within the ischemic retina and reduces pathological neovascularization [53,54]. Some authors have demonstrated the association between elevated serum/plasma TNF-α levels within days after birth and later development of treatment-warranted ROP [28,30]. However, TNF-α also has paradoxical anti-inflammatory properties [55,56]. Some authors discovered low TNF-α levels in the amniotic fluid retrieved during cesarean delivery correlating with the development of ROP in preterm infants [57]. Others found no difference in TNF-α levels in umbilical cord blood between ROP and control groups [58]. These contradictory results may be due to differences in the times and locations of sample collection.

### 2.11. Vascular Endothelial Growth Factor (VEGF)

The VEGF family is one of the key molecules involved in the pathological angiogenic changes in the retina [3,13]. In physiological conditions, they promote embryonic vascular development [59] and possess inner retinal neuroprotective properties [60]. However, over-production of VEGFs can lead to devastating damage to the retina via pathological angiogenesis, abnormal vessel sprouting, and increased vascular permeability [61]. VEGF ligands comprise five members in humans: VEGF-A/B/C/D and placental growth factor (PlGF). They function through three tyrosine kinase cell receptors: VEGFR1, VEGFR2, and VEGFR3 [62]. Each type of ligand has identical interactions with different receptors, as VEGF-A binds to VEGFR1 and 2, VEGF-B and PlGF only bind to VEGFR1, and VEGF-C and VEGF-D primarily bind to VEGFR3 [63]. VEGF signaling via VEGFR-2 is the major pathway in both normal and pathological angiogenesis, regulating endothelial cell migration and survival and promoting endothelial permeability [64,65]. The biological role of VEGFR1 is similar to that of VEGFR2; it plays crucial roles in endothelial cell migration and differentiation, involving the vascular development and regulation of inflammatory cells [66,67,68]. VEGFR1 also binds to VEGF-A. In particular, due to its higher affinity to VEGF and lower tyrosine kinase activity than VEGFR2, it has a decoy effect on VEGF-A [66,67] and may have a therapeutic effect in the amelioration of uncontrolled VEGFR2 activation as well as subsequent neovascularization [61].

VEGF is downregulated in phase I ROP and upregulated by Müller glial cells in the peripheral avascular retina in phase II ROP [69,70,71]. When exposed to a hypoxic environment, hypoxia-inducible factor (HIF)-1α stabilizes and promotes the expression of VEGF, along with other pro-angiogenic factors. Other growth factors, such as erythropoietin (EPO) [72,73,74], IGF-1 [75,76,77,78], TGF-β [79], and FGF [71,80], enhance VEGF signaling [61,81]. Neurons and glial cells also modulate VEGF via the suppressor of cytokine signaling 3 (SOCS3) and signal transducer and activator of transcription 3 (STAT3) signaling pathways [82].

In vitreous samples collected during vitrectomy, VEGF levels were higher in vascularly active ROP eyes than in the non-ROP controls [83,84]. Nonetheless, published data remained inconclusive regarding the systemic levels of VEGF in association with ROP. Among preterm neonates who eventually developed ROP, some researchers found that VEGF concentrations in the early period of life in cord blood at birth and circulating blood are significantly lower than those in their non-ROP counterparts [72,85,86], while some observed the opposite results [87,88,89] and others found no difference [40,58,90].

Other studies have investigated different samples and provided new insights into ROP biomarkers. Vinekar et al. [91] evaluated VEGF levels in tears and observed significantly low levels of VEGF in patients who had no ROP at the initial screening test but later developed ROP. Liang et al. [92] reported that high aqueous levels of VEGF were associated with ROP involving the posterior zone of the retina. As anti-VEGF monoclonal antibodies have become one of the most important treatments for ROP [93], the role of systemic VEGF in the pathogenesis of ROP needs to be explored further.

### 2.12. EPO

EPO, an oxygen-regulated growth factor, plays an important role in the regulation of hematopoiesis [94]. Its additional functions, such as neuroprotection [95,96], anti-apoptosis [97,98], anti-oxidation [99], angiogenesis promotion [100], and BRB maintenance [101], have also been discussed recently. Mainly modulated by HIF-2, EPO receptor (EPOR) signaling can also be enhanced by VEGF-A [102], along with the activation of nitric oxide synthase (NOS) in the pathological angiogenesis of ROP [72,73,74].

In ROP, EPO is regarded as a double-edged sword for its pro-angiogenic effect, which could be beneficial in the first phase of ROP when vaso-obliteration occurs due to hyperoxia, but is harmful in the second phase by aggravating abnormal neovascularization [103,104]. In mouse OIR models, hypoactive EPOR signaling contributes to retinal vascular loss under hyperoxia [73,74,103,104]. In addition, the administration of exogenous EPO could be beneficial in reducing the avascular retinal area [73], especially at an earlier period of life [103,104]. The additional benefit of EPO in preventing photoreceptor cell death has also been discovered [105,106].

Sato et al. [84] reported that among patients who had already developed stage 4 ROP, the vitreous level of EPO was significantly higher in eyes with vascular-active ROP. Many authors have investigated the correlation between ROP occurrence and EPO blood concentrations in the early stage of life in preterm infants but yielded conflicting results: some reported lower EPO levels in association with ROP occurrence [107,108], some reported higher levels in an extremely premature group (GA<28 weeks) [109], while others reported no association [72,110]. Further studies are required to elucidate the true correlation between ROP and blood EPO levels in a time-dependent manner.

### 2.13. Insulin-Like Growth Factor-1 (IGF-1)

IGF-1 is a crucial factor for the normal growth of many tissues and organs [111], and its plasma level rises with fetal GA, particularly during the third trimester of pregnancy [112]. During the normal development of human eyes, its contribution to retinal vascularization has been frequently reported [113,114,115]. Importantly, high IGF-1 concentration is essential to maximize the pro-angiogenic effect of VEGF and to trigger downstream signaling pathways mediated by mitogen-activated protein kinase (MAPK) and Akt, promoting endothelial cell proliferation and integrity maintenance [75,76,77].

IGF-1 also plays a role in stimulating VEGF synthesis [78]. Therefore, preterm infants with low serum levels of IGF-1 in early life are at risk of poor retinal blood vessel growth and a larger area of the avascular retina [116,117]. Accordingly, most studies have found lower blood IGF-1 levels correlating to ROP development and increased ROP severity [28,72,87,116,117,118,119,120]. Exogenous IGF-1 administration even prevented OIR in mouse models [121]. However, based on animal studies, IGF-1 may also contribute to pathological uncontrolled neovascularization in the proliferative stage of ROP [122,123]. In mouse models, suppression of retinal neovascularization has been reported following the administration of an IGF-1 antagonist [124].

### 2.14. Insulin-Like Growth Factor-Binding Proteins (IGFBPs)

IGFBPs are important regulators and serum carriers of IGFs [125]. In addition, IGFBPs possess IGF-independent properties that are involved in cell proliferation, survival, development, growth, and angiogenesis [78]. Among the IGFBPs family, IGFBP-3 has been discussed the most in terms of its association with ROP. In mouse OIR models, IGFBP-3 acts independently of IGF-1, preserving retinal vessels under oxygen-induced damage, facilitating vessel regrowth, and decreasing retinal neovascularization tuft formation [126].

Some authors have reported a significant correlation between higher blood levels of IGFBP-3 and a lower risk of severe ROP in preterm infants [117,127], indicating that IGFBP-3 deficiency may be involved in the pathogenetic process of proliferative ROP. In one study, increased amniotic fluid IGFBP-2 levels correlated with the occurrence of severe ROP [128]. However, others reported no association between ROP development and maternal plasma levels of IGFBP-2 and IGFBP-3 [29], or with cord plasma levels of IGFBP-1 and IGFBP-2 [31].

### 2.15. Transforming Growth Factor (TGF)-β

TGF-β is known for its role in immune modulation, cell growth regulation, and vascular responses [129]. Its potential effects on ocular structures and retinal vessel development have been confirmed in several studies [129,130,131,132,133]. TGF-β has a bipolar effect in different environments, as it can both trigger and inhibit angiogenesis [134]. It promotes endothelial cell proliferation and migration at low concentrations but exhibits an inhibitory effect at high concentrations [129]. In normal physiological conditions, TGF-β and its interaction with VEGF play crucial roles in retinal vessel development and maintaining pericyte integrity [79,129,135]. TGF- β also upregulates the expression of VEGFR-1, protecting retinal vessels from hyperoxia-induced degeneration and inhibiting abnormal neovascularization [79]. However, in OIR models, the bFGF-activated TGF-β1/smad2/3 signaling pathway is over-expressed and induces pathological angiogenesis and damage to photoreceptors [133,135]. In short, both the over-expression and under-expression of TGF-β should be avoided to optimize normal retinal vessel development. Sood et al [27]. reported a significantly lower serum level of TGF-β in preterm infants with type 1 or 2 ROP on postnatal days 7 to 21, suggesting that the lack of vessel protection by TGF-β can lead to oxygen-induced damage.

### 2.16. FGF

FGF is essential for embryonic development in terms of angiogenesis, cell proliferation, and migration [136]. Under hypoxia, FGF2 (or bFGF) derived from RPE elicits its angiogenic effects via both VEGF-dependent and VEGF-independent pathways [133,137,138]. Its signaling directly through the FGF receptor can cause pathologic angiogenesis via STAT3 activation [80]. FGF2 also potentially binds to VEGFR2, triggering downstream VEGF signaling [80]. In addition, it regulates vascularization by activating the TGF-β1 and p-smad2/3 pathways [133]. However, the role of FGF in the pathogenesis of ROP has not been clearly understood. One study reported that FGF2 was not the main characteristic for the development of neovascularization [139]. Instead, it has a neuroprotective effect in photoreceptor cells, helps preserve visual responses, and prevents retinal degeneration in the OIR model [139]. Other studies have shown increased FGF2 expression in the vitreous of infants with ROP at the time of vitrectomy [23] and in the rat OIR model during the neovascularization phase [140]. Holm et al. [39] reported a correlation between elevated serum FGF2 levels in the first three postnatal weeks and the risk of pre-threshold ROP, but Yu et al. [36] found no significant difference in serum FGF2 levels at birth between patients with ROP and non-ROP controls.

### 2.17. Angiopoietin (Ang)

Ang-1 and Ang-2 are growth factors that work in concert with VEGF and contribute to physiological and pathological neovascularization [141]. Both Ang-1 and Ang-2 function through the Tie2 receptor tyrosine kinase, although they exert agonizing (Ang-1) and antagonizing (Ang-2) effects, respectively [142]. Tie2 is phosphorylated upon activation by Ang-1, fires the downstream Akt and ERK cellular pathways, and stabilizes vascular integrity [143]. In the meantime, the expression of NOS maintains endothelial cell survival, and the downregulation of nuclear factor-kappa B (NF-κB) alleviates inflammation [144]. Ang-2 induced by both hypoxia and VEGF competes against Ang-1, dephosphorylates Tie2, and initiates neovascularization [145]. The balance between vitreous Ang-1 and Ang-2 levels has been proven to be crucial in retinal vessel development [141]. In patients with ROP, a higher concentration of Ang-2 was observed in the vitreous [141] and the fibroproliferative membranes removed from eyes with stage 5 ROP [146]. Inhibition of Tie2 and VEGF together appeared to be more effective in treating retinal pathological angiogenesis than VEGF inhibition alone [147,148]. However, there is a lack of data regarding systemic Ang levels in patients with ROP.

### 2.18. Platelet-Derived Growth Factor (PDGF)

The PDGF family is a regulator of retinal angiogenesis. However, each isoform has a niche for vessel formation. In ocular tissues, PDGF-BB is the predominant isoform [149]. PDGF-B helps to recruit pericytes and vascular smooth muscles [150]. PDGF deficiency in rat OIR causes pericyte degeneration, resulting in abnormally dilated vessels with hemorrhages and, most importantly, elevated VEGF/VEGFR-2 expression with subsequent angiogenesis [151]. PDGF-CC, another isoform, intensifies MMP-2 and MMP-9 expression and augments monocyte migration [152]. Inhibition of PDGF-CC or PDGF-DD reduced both choroidal and retinal neovascularization in animal models [153,154]. In one human study, preterm infants with low platelet counts and low serum levels of PDGF-BB at 32 weeks of PMA were at risk of severe ROP [85]. Some researchers found that the combination of anti-VEGF therapy with anti-PDGF resulted in better final vision compared to anti-VEGF therapy alone in other proliferative retinopathy [155]. Still, such a therapy is yet to be tested in ROP.

### 2.19. Endoglin

Endoglin is a transforming growth factor-beta (TGF-β) auxiliary co-receptor that is highly expressed in the angiogenic endothelial cells of embryos, inflamed tissues, and tumors [156]. Soluble endoglin (sEng, circulating form) displays anti-angiogenic activity by downregulating TGF-β signaling [157,158]. Higher vitreous sEng levels were detected in patients with proliferative diabetic retinopathy, implicating its role in impairing normal retinal vascular development [159]. Accordingly, elevated amniotic fluid sEng levels were found to be significantly related to the development of ROP requiring treatment [128].

### 2.20. Endostatin

Endostatin is an anti-angiogenic protein that inhibits endothelial cell proliferation, migration, invasion, capillary tube formation, and retinal VEGF secretion [160]. In a mouse OIR model, the level of endostatin-like protein varied reciprocally to the VEGF level and presumably took part in the regression of vessels [161]. Elevated endostatin levels in amniotic fluid have been shown to be correlated with the development of severe ROP in preterm infants [128] possibly highlighting its role in microvascular degeneration during phase 1 ROP. Some authors have shown that the retinal administration of endostatin is effective in preventing pathologic retinal neovascularization in an OIR model [162].

### 2.21. Endocan

Endocan or endothelial cell-specific molecule-1, a soluble proteoglycan mainly produced by vascular endothelial cells, actively mediates cell adhesion, migration, proliferation, and neovascularization [163]. It is essential for the pathogenesis of vascular disorders, inflammation, and endothelial dysfunction [164]. In mouse OIR models, retinal endocan levels are significantly upregulated at critical time points [165]. In vitro, under hypoxic conditions, human retinal microvascular endothelial cells show increased endocan expression, promoting tube formation and vessel sprouting [165]. Some authors have even considered serum endocan as a marker for predicting severe ROP [87].

### 2.22. Neurotrophins

Neurotrophins are a family of growth factors that promote the survival, development, and function of neurons in the central and peripheral nervous systems [166,167]. Brain-derived neurotrophic factor (BDNF), one of the neurotrophins, is important in both the functional and structural development of the inner retina through BDNF/TrkB signaling [168,169]. In mouse OIR models, the hyperoxia-exposed group had significantly lower BDNF expression than the control group reared in room air [170]. BDNF exerted a protective effect by stabilizing the retinal vasculature [170]. The lack of BDNF expression causes endothelial cell apoptosis and damages cell–cell connections [170]. A consistent result was demonstrated as lower levels of serum circulating BDNF, along with neurotrophin 4, were observed in ROP infants compared to their non-ROP counterparts [27,85,171,172]. Several studies focused on the genetic contributions of BDNF to the development of ROP [173,174,175]. Certain polymorphisms of BDNF gene polymorphisms have been shown to be associated with severe ROP [175]. Although the role of neurotrophins in ROP development is not fully understood, there may be a connection between neurovascular interactions in the retina and the pathogenesis of ROP.

### 2.23. Stromal-Derived Factor 1α (SDF-1α)

SDF-1α, a chemokine mediated by HIF-1, is up-regulated in ischemic tissues [176,177]. The expression of SDF-1α and its membrane receptor on endothelial cells, CXCR4, are both enhanced by VEGF and bFGF [178]. SDF1/CXCR4 signaling triggers VEGF secretion by endothelial cells, promoting endothelial progenitor cell trafficking, cell migration, and angiogenesis [177,179,180]. Sonmez et al. [83] revealed elevated vitreous SDF-1α levels in eyes with vascularly active stage 4 ROP. In OIR models, increased immunolabelling of SDF-1 in endothelial cells and strong expression of CXCR4 in Müller cells have also been demonstrated [181], and inhibition of SDF-1α was presumed to be associated with reduced pathological neovascularization [182,183].

### 2.24. RANTES

RANTES is a chemokine that contributes to innate immunity and is particularly important in the neonatal period [184]. Although mainly expressed by T lymphocytes and monocytes [185], RANTES is also secreted by retinal endothelial and pigment epithelial cells to initiate inflammation [186,187]. Sood et al. [27] discovered that RANTES blood levels on postnatal days 7-21 were lower with increasing ROP severity. On the contrary, vitreous RANTES levels were significantly elevated in both vascularly active and inactive ROP eyes than in non-ROP controls at the time of vitrectomy [23]. This discrepancy may be explained by the different stages of ROP at which the samples were retrieved. However, the exact underlying mechanism requires further investigation.

### 2.25. MCP-1

MCP-1 is a chemokine mainly produced by microglia, which further induces the migration and aggregation of microglia and/or circulating monocytes through the BRB in ischemic retinas [188,189]. MCP-1 enhances neovascularization by acting as a direct angiogenic factor itself [190], or by recruiting pro-angiogenic macrophages [191]. Cord blood MCP-1 levels were higher in neonates who developed ROP than in their non-ROP counterparts [36]. On postnatal day 3, prolonged supplemental oxygen exposure was associated with higher blood MCP-1 concentrations in extremely low BW neonates (<1000 g) [192]. However, in one study, vitreous MCP-1 levels did not differ significantly among vascularly active ROP eyes, vascularly inactive ROP eyes, and control eyes at the time of vitrectomy [23].

### 2.26. Macrophage Inflammatory Protein-1 (MIP-1)

MIP-1, a microglia-derived chemokine similar to MCP-1, is involved in post-ischemic inflammation as a chemoattractant for macrophages in the hypoxic retina [188]. In mouse OIR models, MIP-1β was found to be the most upregulated gene under hypoxia [193] and potentially gave rise to the physiological revascularization of the avascular retinal area [194]. Yu et al. [36] found that preterm infants with ROP displayed significantly higher cord blood MIP-1β levels than their healthy counterparts. MIP-1β, along with MCP-1, was negatively correlated with GA and BW in neonates with ROP [36]. In another study, however, neither MIP-1α nor MIP-1β vitreous levels differed significantly between eyes with and without ROP [23].

### 2.27. Eotaxin

Eotaxin is a chemokine secreted by a variety of cells, including macrophages, eosinophils, endothelial cells, and fibroblasts. It primarily attracts eosinophils, mediates inflammation, and promotes angiogenesis via the receptor CCR3 [195,196]. Its expression in the vitreous, neurosensory retina, or choroid has been implicated in the pathogenesis of AMD [196], proliferative diabetic retinopathy, and choroidal neovascularization [197]. Likewise, vitreous eotaxin concentrations were significantly higher in ROP eyes than in non-ROP controls [23]. Nonetheless, a study revealed lower plasma eotaxin levels in the early neonatal period of premature infants being associated with clinically significant ROP [198], possibly reflecting phase 1 disease characterized by retinal vaso-obliteration. The biphasic pattern of eotaxin was also observed in a mouse OIR model, and the anti-CCR3 antibody was proven to be efficacious in suppressing pathological neovascularization [199].

### 2.28. Interferon (IFN)-γ

IFN-γ, secreted mainly by T lymphocytes and natural killer cells, is a potent macrophage activator [200]. IFN-γ exhibits pro-inflammatory, antiviral, anti-proliferative, pro-apoptotic, and antitumor properties [201,202]. In vivo, IFN-γ impedes the proliferation, migration, and tube formation of endothelial cells via STAT1 signaling, even under VEGFA-treated conditions [203]. In a mouse OIR model, IFN-γ displays prominent anti-angiogenic effects [203]. Although IFN-γ levels in cord blood and vitreous were found to be comparable in neonates with and without ROP [23,58], one study observed that IFN-γ levels in the aqueous were significantly higher in threshold ROP eyes than in pre-threshold ROP eyes before intravitreal anti-VEGF treatment and non-ROP eyes at the time of congenital cataract extraction [204]. Whether this elevation represents advanced inflammation in ROP eyes or a compensatory effect against pathological neovascularization warrants further investigation.

### 2.29. Granulocyte Colony-Stimulating Factor (G-CSF)

G-CSF is a regulator of hematopoiesis and immunity [205]. In addition to its effect on neutrophil stimulation, it also triggers angiogenesis in ischemic tissue [206] and participates in the synthesis of IGF-1 [207]. Its neuroprotective effect against ischemic injury in the inner retinal layer has been documented in several studies [208,209,210]. The mechanism involves the STAT3 and PI3K/AKT pathways and shows potential therapeutic effects on retinal ischemic disorders [208,209,210].

A significant increase in vitreous G-CSF levels was noted in infants with ROP compared to their non-ROP counterparts [23,49]. Nonetheless, the serum G-CSF levels at birth did not differ significantly between patients with ROP and non-ROP controls [36]. In OIR models, the administration of G-CSF showed a significant benefit in reducing oxygen-induced retinal vascular obliteration and protecting the retina from hyperoxia-induced apoptosis and both structural and functional damage [211]. Practically, however, its therapeutic effect requires further investigation. In humans, one study revealed an insignificantly lower incidence of threshold ROP in infants administered G-CSF than in those not treated [212].

The characteristics and actions of all the above-mentioned cytokines in ROP are summarized in Table 1. Evidence is often limited by the small sample size, non-replicability, or conflicting results of different studies, possibly attributed to the varying basic characteristics (i.e., degree of prematurity, ethnicity) of study samples, protocols of perinatal care, and timing and source of sample retrieval in different institutions. Although BRB breakdown occurs in ROP pathology [213], the display of cytokines frequently differs in the blood and te vitreous. A specific cytokine may exert both pro- and anti-inflammatory and pro- and anti-angiogenic properties, depending on the dosage, timing, duration, and target tissue. All of these factors should be considered when interpreting the results of these studies.

## 3. Inflammatory Pathways and Cellular Responses

The inflammatory and angiogenic pathways involved in the two phases of ROP development are shown in Figure 1. Preterm neonates encounter oxidative stress from all sources, such as light exposure, infection, ischemia/reperfusion-induced inflammation, long-term parenteral nutrition, blood transfusions, increased levels of non-protein-bound iron, and most importantly, high supplemental oxygen [4]. Their immature antioxidant systems and long-chain polyunsaturated fatty acid (LC-PUFA)-rich retinas make them susceptible to oxidative damage [214]. Oxidative and inflammatory pathways share some common signaling molecules. Under hyperoxia (phase I ROP), pro-angiogenic factors, including VEGF, IGF-1, and EPO, are suppressed, leading to retinal vaso-obliteration [215]. Tissue injury caused by the overproduction of reactive oxygen species (ROS) results in the activation of immune cells, mainly microglial cells, which are resident macrophages in the retina [216]. IL-1β, TNF-α, and IL-6, the primary cytokines at play in the initial stage of inflammation, further cause deleterious effects in the retina [20] and aggravate retinal microvascular degeneration [18].

Subsequently, inadequate perfusion of the developing retina leads to the development of phase 2 disease [217]. HIF, a redox-sensitive transcription factor, is stabilized under low-oxygen conditions due to the inhibition of its hydrolytic enzymes, the prolyl hydroxylase domain (PHD) proteins [218]. HIF induces growth factors involved in angiogenesis, such as VEGF, EPO, PDGF, and Ang2, and enhances endothelial cell proliferation, migration, and tube formation [219,220,221]. In ischemic retinal tissue, ROS generated through NADPH oxidase [222] further causes the release of various aforementioned cytokines in the retina via upregulation of the pro-inflammatory signaling pathways mediated by NF-κB, protein kinase C, and MAPK [223]. Chemokines, such as IL-8 and MCP-1, mediate the recruitment of more immune cells [37]. Some anti-inflammatory cytokines, such as IL-10 and IL-19, have been reported to display pro-inflammatory properties [42]. Inflammatory cytokines, growth factors, and ROS stimulate the secretion of MMP-2 and MMP-9 by RPE, which degrades the extracellular matrix, induces endothelial cell migration, and recruits more growth factors [224,225]. Inflammation superimposes overactive VEGF signaling which eventually disorients endothelial cell divisions and allows vessel growth to extend into the vitreous rather than staying within the retina [226,227,228].

## 4. Correlation with Other Maternal or Neonatal Diseases

The alteration of systemic cytokines resulting from all kinds of perinatal and neonatal inflammation potentially contributes to the pathogenesis of ROP (Figure 1) [27]. For this purpose, understanding the role of systemic cytokines in both maternal and neonatal diseases may provide insights into the pathophysiology of ROP.

### 4.1. Pre-Eclampsia

Preeclampsia is a pregnancy-specific disease presenting with maternal hypertension and various organ failures and is associated with multiple neonatal morbidities [229]. The pathogenesis of preeclampsia involves soluble fms-like tyrosine kinase-1 (sFlt-1), also known as VEGFR1 [230], which attenuates VEGFR2 and has anti-angiogenic properties [61]. Clinical studies have reported increased serum sFlt-1 concentrations in newborns exposed to preeclampsia within the first three days of life [230,231]. However, whether pregnancy-induced hypertension and pre-eclampsia are risk factors for ROP remains inconclusive [1]. One recent meta-analysis reported no significant association between ROP and preeclampsia, but a high heterogeneity among the included studies was shown [232]. Some researchers demonstrated larger retinal avascular areas and more severe ROP at first screening in preterm infants born to hypertensive mothers, but no association was revealed between gestational hypertension and the worst severity of ROP in further examinations [233]. On the contrary, others showed a reduced risk of ROP in preterm infants born to pre-eclampsia mothers [234,235] possibly due to the anti-angiogenic effect of sFLt-1 [234]. Further studies are required to elucidate the true correlation between pre-eclampsia and ROP.

### 4.2. Maternal Diabetes Mellitus (DM)

Some researchers have reported maternal DM as a risk factor for ROP development [236,237,238], while others claimed no relationship [239,240,241]. One recent meta-analysis showed no association between maternal DM and ROP [242], but the heterogeneity among studies in terms of baseline subject characteristics and diabetes management may be overlooked. Concentrations of growth factors, such as IGFBP1 and IGFBP2, in cord blood were found to be significantly lower in neonates born to women with gestational DM than in controls [243]. However, other growth factors, such as IGF-1, IGFBP3, Ang-2, VEGF, and PlGF, did not differ significantly [243,244].

### 4.3. Chorioamnionitis

A recent meta-analysis concluded that maternal chorioamnionitis increased the risk of ROP in preterm neonates [245]. Infants exposed to chorioamnionitis more frequently develop postnatal infections and inflammation, and the resultant oxidative stress makes them vulnerable to ROP [246,247]. Both maternal inflammatory response and the subsequent fetal inflammatory response syndrome promote systemic pro-inflammatory cytokines in neonates, including TNF-*α*, IL-1, IL-6, and IL-8, especially within the first 72 hours of life [30,31,246,248,249,250,251,252]. In addition, maternal systemic inflammation also causes decreased fetal IGF-1 [246]. The display of these cytokines, attributed to ROP development, is mentioned.

### 4.4. Respiratory Distress Syndrome (RDS)

RDS in preterm infants typically progresses over the first two–three days of life and resolves by one week of age with an increased production of endogenous surfactant [253]. Preterm infants with RDS often require mechanical ventilation and supplemental oxygen, both of which are widely recognized risk factors for ROP [1,254,255,256,257,258,259]. In addition, the alternation in cytokines of neonates with RDS compared to non-RDS counterparts may also take part. Higher IL-6, IL-10, and lower IL-12 levels in cord blood [260] and lower plasma VEGF levels during the first week of life [261] have been observed in infants with RDS.

### 4.5. Patent Ductus Arteriosus (PDA)

PDA refers to the failure of the ductus arteriosus to close completely 1–2 days after birth [262]. Prenatal and postnatal infections and the resultant inflammation process have been proposed to contribute greatly to PDA [263]. Accordingly, elevated levels of cytokines, such as IL-6, IL-8, IL-10, IL-12, growth/differentiation factor 15, MCP-1, and MIP-1α, in cord blood have been reported to be related to the development and persistence of PDA [264,265]. Many studies demonstrated PDA as an independent risk factor for ROP development [266,267,268,269,270]. Inflammation is supposedly a confounding factor contributing to both PDA and ROP development. Some researchers have claimed that the use of non-steroidal anti-inflammatory drugs, such as indomethacin, to manage PDA gives rise to increased VEGF levels in the eyes and subsequent retinal neovascularization [271,272], while others disagree [273]. Mitsiakos et al. [270] also indicated the potential etiology of alterations in retinal perfusion resulting from open ductus arteriosus. Further studies are required to elucidate the association between PDA and ROP.

### 4.6. Intraventricular Hemorrhage (IVH)

IVH is a serious complication in preterm infants, and its association with ROP has been demonstrated in various studies [254,274,275,276,277,278,279]. IVH usually occurs within the first week of life, causing poor development of neurons and glial cells [280]. While the pathogenesis of IVH is considered to be multifactorial, the destruction of germinal matrix vessels is an important mechanism [281]. The subsequent inflammatory response caused by the hemorrhage results in further damage to the adjacent tissue [282,283]. In addition to the local elevation of pro-inflammatory mediator levels in the cerebrospinal fluid (CSF) [283,284] and periventricular tissue [285], systemic changes, including elevated IL-1β, IL-6, TNF-α, and EPO levels in cord blood [286,287,288], and elevated IL-6, IL-8, and MCP-1 levels in circulating blood were observed [289,290]. Variants of polymorphic genes regulating the expression of cytokines, such as IL-1β and TNF-α, were also reported in patients suffering from severe IVH [291].

VEGF and TGF-β, which are crucial for the development of ROP, also participate in the pathogenesis of IVH. Locally high VEGF and Ang-2 levels in the germinal matrix facilitate angiogenesis, and the combination of low TGF-β expression and glial fibrillary acidic protein deficiency leads to fragile vessels in the germinal matrix [281]. Shimi et al. [292] evaluated both blood and CSF levels of VEGF on the 1st and 3rd days of life and reported significantly higher levels in patients who later developed IVH. However, the cytokines that play a role in both IVH and ROP require further investigation.

### 4.7. Bronchopulmonary Dysplasia (BPD)

Clinically, BPD is defined as the need for supplemental oxygen either at 4 weeks postnatal age or 36 weeks PMA [293], although lung injury commences soon after birth [294]. BPD is characterized by chronic lung disease mainly due to prolonged application of mechanical ventilation and oxygen toxicity after birth [294] and is identified as an independent risk factor for ROP [27,279,295]. In spite of the contribution from exposure to a high-oxygen environment, cytokines level alternation in BPD patients was described in several studies and may also play a role in the pathogenesis of ROP. Higher blood levels of IL-6 and IL-8 within the first month of life are associated with BPD and increased severity [261,296]. Increased serum levels of TNF-α in the postnatal 4th week were also shown to be a risk factor for BPD [296]. Other studies have proposed impaired angiogenesis as a mechanism of BPD, and endostatin and PlGF levels were elevated in the cord plasma of patients with BPD [297,298].

### 4.8. Anemia and Red Blood Cell (RBC) Transfusion

Preterm infants with low hemoglobin (Hb) levels are considered to be at risk for ROP [299,300]. As Hb delivers oxygen throughout the whole body, a decreased concentration of circulating hemoglobin represents the general hypoxia status and potentially increases VEGF secretion [301]. Several studies have also pointed out the association between ROP development and transfusion of RBCs, especially in the early postnatal period, when phase 1 of ROP occurs [302,303,304]. The risk of ROP is even higher as the frequency and volume of transfusions increase [305]. The proposed hypothesis of etiology included (1) transfusion of adult RBCs, which is mainly composed of HbA that releases more oxygen than HbF, the primary type of Hb in infants, to the retinal tissue and causes hyperoxia-induced damage [302,305]; (2) accumulation of free iron leading to oxidative stress [302,306]; and (3) alteration in cytokine profiles after blood transfusion [307,308,309]. In patients receiving a massive RBC transfusion, higher blood IL-1β, IL-6, IL-8, IL-17A, and TNF-α levels were detected later in life [307,308,309].

### 4.9. Thrombocytopenia and Platelet Transfusion

Platelets are responsible for the storage, transport, and release of abundant growth factors, such as IGF-1, IGFBP3, VEGF-A, PDGF, and BDNF, from their α-granules [85,310,311]. In rodent models, platelets showed a significant anti-angiogenic effect on retinal endothelial cells by reducing VEGF-A production [312]. The correlation between thrombocytopenia and severe ROP has been hypothesized and proven in several studies [85], especially during the first week of life [313,314]. Contrary to the theoretical protective effect of platelets, transfusion of platelets was reported to be an aggravating factor for ROP development due to the contained pro-inflammatory cytokines and their interactions with immune cells [302,305,315].

### 4.10. Fresh-Frozen Plasma Transfusion

Dani et al. [316] reported that among infants with a GA < 29 weeks, those who received two or more transfusions of fresh-frozen plasma in the first week of life had a lower risk of developing ROP. High concentrations of IGF-1 and IGFBP are thought to contribute to this result.

### 4.11. Necrotizing Enterocolitis (NEC)

NEC is a devastating inflammatory bowel disease induced by innate immune responses against the gut microbiota [317]. It typically occurs in the second to the third week of life after starting oral intake [317]. Published data have consistently revealed a correlation between NEC and ROP development [1,318,319]. On postnatal day 1, significantly lower circulating TGF-β1 and IL-2 and increased IL-8 levels were found in neonates who later developed NEC compared to the non-NEC controls [320,321]. Later on, significantly higher levels of other cytokines, such as IL-1β, IL-6, IL-10, IL-1Ra, MCP-1, and MIP-1β, were also reported in these infants [321,322,323]. During the first 3 weeks of life, systemic IL-1β, IL-2, IL-6, and IL-10 levels showed a decreasing trend over time, whereas IL-18, MIP-1β, and TGF-β1 levels increased with postnatal age [321]. The alternation in these cytokine levels may be potential biomarkers for NEC diagnosis and may take part in the development of ROP.

### 4.12. Sepsis and Fetal Inflammatory Response Syndrome

Sepsis, which may develop at any time in the neonatal period, is a well-known independent risk factor for ROP [236,279,324,325]. The resultant hemodynamic instability causes fluctuation in blood oxygen saturation, which might lead to retinal ischemia [326]. In addition, sepsis, along with fetal inflammatory response syndrome, causes systemic elevation of the levels of various pro-inflammatory cytokines, including IL1-β, IL-6, IL-10, and TNF-α, which further aggravates ROP severity [326,327,328].

## 5. Novel Pharmacological Agents Linking to Cytokine Signaling Pathways

### 5.1. Current Prophylactic Treatments for ROP

#### 5.1.1. EPO and Derivatives

Anemia of prematurity and blood transfusions are both potential risk factors for ROP [1] that can be avoided with EPO derivative supplementation [329]. The pro-angiogenic properties of EPO itself may reduce vaso-obliteration in phase 1 ROP but aggravate neovascularization in phase 2 ROP [103,104]. Therefore, there has been a large debate on whether exogenous EPO can prevent ROP, and the timing of administration probably plays a crucial role. An RCT that evaluated the effect of early prophylactic EPO (within 72 h of birth and then once every other day for 2 weeks) specifically on ROP revealed benefits in infant boys or in infants with GA > 28 weeks and BW > 1500 g [330]. In contrast, in a recent Cochrane meta-analysis comprising two high-quality RCTs, early EPO (initiated at age < 8 days) for preventing blood transfusion in preterm infants was related to a significantly increased risk of any grade ROP compared to late EPO (initiated at 8–28 days of age) [331]. However, in two meta-analyses enrolling more RCTs/quasi-RCTs that compared early or late EPO with placebo or no intervention, late EPO brought about a trend of increased risk for ROP [332], while early EPO had no impact [333]. EPO administration does not seem to be promising for managing ROP based on current evidence.

#### 5.1.2. PUFA Supplements

PUFAs, such as docosahexaenoic acid (DHA) and arachidonic acid (AA), constitute the fundamental structure of neurons and endothelial cells [334]. They contribute greatly to retinal physiological functions, specifically cell signaling mechanisms involved in phototransduction [335]. DHA is a ω-3 PUFA, whereas AA is a ω-6 PUFA. Substantial ω-3 and ω-6 long-chain PUFAs delivered from the mother through the placenta during the third trimester of gestation are unavailable to preterm infants [336,337].

In a mouse OIR model, dietary ω-3 PUFAs reduced the area of the avascular retina and alleviated pathologic neovascularization, partly through the suppression of TNF-α in a subset of microglia closely associated with retinal vessels [54]. In light of this, a meta-analysis showed that long-chain PUFA supplementation in infant formulas improved visual acuity up to 12 months of age [338]. Pawlik and al. demonstrated that parenteral ω-3 supplementation decreased the risk of severe ROP in very preterm infants [339]. In another RCT (NCT03201588), daily enteral oil supplementation with AA and DHA from postnatal day 3 to PMA week 40 reduced the occurrence of severe ROP [340]. However, the authors indicated that DHA was only effective in preventing severe ROP in infants with sufficiently high serum AA levels [341]. Another RCT (NCT02683317) showed that enteral DHA supplementation for 2 weeks was able to reduce the risk of stage 3 ROP in preterm infants [342]. Further research is warranted to determine the optimal composition of PUFAs, route of administration, and duration of treatment for preventing ROP.

### 5.2. Current Curative Treatment Strategy for ROP

#### Anti-VEGF

Inhibition of VEGF–VEGFR signaling has recently become an established treatment option for intravitreal neovascularization. Bevacizumab, a humanized anti-VEGFA monoclonal antibody initially approved for cancer therapy, was the first drug reported for ROP treatment [8]. Compared to conventional laser therapy, the BEAT-ROP trial proved that intravitreal bevacizumab could decrease the risk of reactivation before 54 weeks of PMA by five times in zone I ROP [8]. Nonetheless, some small-scale randomized controlled trials (RCTs) have yielded contradictory results regarding reactivation rates compared to laser therapy [343,344]. Notably, bevacizumab was superior to laser therapy by allowing for continued normal vessel growth into the peripheral retina [8].

Introduced by the RAINBOW trial, ranibizumab is a monoclonal antibody Fab fragment neutralizing VEGF-A that showed an advantage over laser therapy in managing ROP with a 24-week safety profile and fewer unfavorable ocular outcomes, such as high myopia [7,345]. Aflibercept is a soluble fusion protein that not only binds to VEGF-A but also to VEGF-B and PlGF [346,347]. Chen et al. [348] found that aflibercept was effective and well tolerated for the treatment of Type 1 ROP, but Ekinci et al. [349] demonstrated that more re-treatments were needed compared to the primary laser. FIREFLYE (NCT04004208) and BUTTERFLYE (NCT04101721) are two ongoing phase 3 RCTs comparing intravitreal aflibercept with laser therapy for treating ROP.

Conbercept, a fusion protein that blocks both VEGF-A and VEGF-B, has a 50-fold higher binding affinity than bevacizumab [350]. Its efficacy in safely resolving ROP has recently been demonstrated [350,351]. Brolucizumab is a newly developed humanized single-chain antibody fragment inhibitor of VEGF-A recently approved for managing adult retinal diseases [352]. Faricimab is a bispecific antibody targeting both VEGF-A and Ang-2, which has recently gained approval from the FDA for managing exudative AMD [353]. The potential efficacy of brolucizumab and faricimab in treating ROP has yet to be confirmed. All the above-mentioned anti-VEGF agents are currently off-label for ROP, except ranibizumab, which has become the first licensed drug in the European Union since 2019 [354].

Anti-VEGF agents have some drawbacks regarding their local side effects, such as RPE degeneration, which may negatively affect the viability of photoreceptors, choriocapillaris, and Müller cell signaling [10]. Moreover, there have been concerns regarding their systemic toxicity retarding organogenesis, particularly neurodevelopmental delay [11,12,13,14]. However, patients receiving lower doses of anti-VEGF often require re-treatment [355]. The optimal treatment probably requires striking a balance between reduced dosage and the risk of late reactivation. Our previous study reported a significant drop in serum VEGF levels up to 2 months in infants with type 1 ROP who received intravitreal bevacizumab but not in those receiving ranibizumab [356]. This result may be attributed to the smaller molecule of ranibizumab, which is eliminated from the bloodstream faster and potentially causes less systemic toxicity [357].

### 5.3. Evolving Treatment Strategies for ROP

#### 5.3.1. HIF–PHD Inhibitors and HIF Inhibitors

HIF is a potential therapeutic target because of its crucial role in stimulating angiogenic factors, as described above. In mouse OIR models, HIF stabilization by PHD inhibitors, such as dimethyloxalylglycinesuch [358], roxadustat, and AR0 [359], in phase 1 ROP prevented retinal vessel dropout and subsequent pathological angiogenesis. The researchers also indicated two mechanisms of action of roxadustat in preventing OIR: direct retinal HIF stabilization and triggering enzymes for aerobic glycolysis, and indirect hepatic HIF-1 stabilization and upregulation of serum angiogenic hepatokines [360]. Singh et al. [361] further highlighted the remote protection of the retina by hepatic HIF-1 via the control of serine/one-carbon metabolism. Interestingly, roxadustat is capable of treating anemia [362], a potential risk factor for ROP [1]. Whether its role in regulating hemopoiesis contributes to its therapeutic effects warrants further investigation.

Conversely, HIF inhibition in phase 2 ROP appears to be a reasonable strategy to directly suppress the pathological action of pro-angiogenic factors. Several HIF inhibitors with varying mechanisms, including topotecan, doxorubicin, a marine fish-derived component, and CITED2 protein-derived peptide, have been studied in OIR models [363,364,365]. Topotecan impedes the accumulation of HIF-1α protein but not mRNA expression [363]. Doxorubicin hampers the binding of HIF-α to hypoxia response elements [363]. Ingredients from *Decapterus tabl*, a marine fish, impede the expression of HIF target genes in a way currently unknown [364]. A peptide derived from the intrinsically disordered protein CITED2 is an endogenous negative feedback regulator of HIF-1α [365]. The combination of this peptide HIF inhibitor with a reduced dose of anti-VEGF aflibercept decreased retinal neovascularization in phase 2 disease and vaso-obliteration in phase 1 disease, which could not be achieved by aflibercept alone [365]. Apurinic/apyrimidinic endonuclease 1/reduction-oxidation factor 1 (APE1/Ref-1) is a multifunctional protein that acts upstream of HIF transcription [366]. APE1/Ref-1 inhibitors are thus considered a potential treatment option for ROP, along with other retinal neovascular diseases [366].

#### 5.3.2. IGF-1/IGFBP-3 Complex

Owing to the aforementioned promising results of preventing mouse OIR [121,126], there is particular interest in IGF-1 and IGFBP-3 administration to reduce the risk of ROP development. Dani et al. [316] reported that preterm infants who received two or more transfusions of fresh-frozen plasma in the first week of life had a lower risk of ROP occurrence and that high concentrations of IGF-1 and IGFBP were thought to be responsible for the result. In one phase 2 RCT (NCT01096784), however, the recombinant human IGF-1/IGFBP-3 complex protected against BPD but not ROP [367]. Dosage optimization may be required for a more optimistic result [367]. A larger-scale phase 2b RCT (NCT03253263) is underway to assess the efficacy of the recombinant human IGF-1/IGFBP-3 complex in the prevention of diseases related to prematurity [368].

#### 5.3.3. Anti-Secretogranin III (Scg3)

Scg3 belongs to the granin family and primarily regulates the biogenesis of secretory granules and the secretion of hormone peptides in endocrine and neuroendocrine cells [369]. It has also been characterized as a disease-associated angiogenic factor that acts through VEGF-independent signaling pathways [369]. Different from VEGF, Scg3 does not bind to healthy retinal vessels. Therefore, in mouse OIR models, anti-Scg3 monoclonal antibodies selectively block pathological angiogenesis but do not hinder normal retinal vascularization as anti-VEGF may [370,371,372]. Moreover, they avoid systemic side effects, possibly caused by anti-VEGF agents, such as renal tubular injury, abnormalities in kidney vessel development, and retarded body weight gain in neonatal mice [370]. Therefore, anti-Scg3 therapy with a wider therapeutic window than anti-VEGF agents offers new hope for the safe and effective treatment of ROP.

#### 5.3.4. Gut Microbiota Modulation

The gut microbiota are microorganisms that reside in the gastrointestinal tract. Disruption of the gut microbiota not only causes gastrointestinal diseases but also exerts systemic effects by changing intestinal permeability, releasing metabolic endotoxins into the blood circulation, inducing systemic inflammatory responses, and altering body growth [373,374]. Systemic inflammatory status potentially takes part in the development of diseases in all systems, including ophthalmic disorders [375]. The concepts of gut–brain [373,376], gut–respiratory [376], and gut–retina axes [374,377] have been discussed in the literature. In mouse models, gut dysbiosis increases the production of serum cytokines, including IL-6, IL-1β, TNF-α, and VEGF-A, which may aggravate pathological angiogenesis in the eye [378]. Another study of mice colonized with fecal samples from poor-growth preterm infants showed lower circulating levels of both IGF-1 and IGFBP-3 compared to those in infants with good growth [373].

Interactions among dietary carbohydrates, gut microbial metabolites, and AMD features in the human eye have been reported [379]. A large cohort study indicated lower diversity of gut microbiota and significant enrichment of *Staphylococcus* in patients with ROP, highlighting the importance of promoting healthy microbiome development in preterm neonates [380]. Skondra et al. [377] demonstrated the enrichment of *Enterobacteriaceae* species with less amino acid biosynthesis in patients with ROP. However, a meta-analysis reported that probiotic supplementation had no benefit in preventing ROP [381]. Therefore, further studies are required to clarify the therapeutic effects of gut microbiota replacement in patients with ROP.

#### 5.3.5. Non-Coding RNAs (ncRNAs)

NcRNAs are functional RNAs that are not translated into proteins and regulate various diseases, including retinal diseases [382]. Based on their molecular weight, they are classified into microRNAs (miRNAs), long non-coding RNAs (lncRNAs), and circular RNAs (circRNAs) [383]. MiRNAs, short ncRNAs with a length of approximately 22 nucleotides, can bind with messenger RNAs (mRNAs) to degrade them or interrupt transcription. LncRNAs with a length >200 nucleotides can modulate gene expression, modify chromatin, or act as miRNA sponges. CircRNAs, which are variable in length, often function as miRNA sponges and can recover mRNA processing or directly participate in transcription [384].

MiRNAs are the most extensively explored ncRNAs in ROP. MiR-18a-5p is upregulated in mouse OIR. Intravitreal agomiR-18a-5p, an miR-18a-5p mimic, regulates pathological neovascularization by targeting HIF-1α and FGF1 [385]. In rat OIR, miR-34a is downregulated. Administration of miR-34a diminishes neovascularization via inhibition of the Notch1 pathway [386]. Desjarlais et al. [387] demonstrated the downregulation of miR-96 in another rat OIR model. Intravitreal injection of the miR-96 mimic before hyperoxia markedly prevented vessel dropout by stimulating pro-angiogenic factors, including VEGF, Ang-2, and FGF2. These factors can be blocked by antagomiR-96. Our previous study reported that intravitreal miR-126 mimic and plasmid effectively inhibit retinal neovascularization by downregulating VEGF-A expression in rat OIR [388]. In another mouse OIR, miR-145 regulated tropomodulin 3 and modified the structure of actin and endothelial cells to enhance pathological neovascularization. Intravitreal miR-145 inhibitors can reduce neovascular areas [389]. Liu et al. [390] demonstrated a reduction in miR-150 expression in the mouse OIR. MiR-150 exhibits anti-angiogenic properties by inhibiting C-X-C chemokine receptor type 4, delta-like ligand 4, and frizzled-class receptor 4 [390]. In another mouse model of OIR, miR-181a-5p was found to inhibit retinal neovascularization via endocrine suppression [391]. Li et al. [392] found that miR-182-5p is downregulated in mouse OIR. The introduction of the miR-182-5p mimic can impede ANG and BDNF expression, thereby reducing cell migration and increasing cell viability and tube formation. MiRNA-223 upregulates the anti-inflammatory genes, *YM1/2* and *IL-4RA*, and downregulates the pro-inflammatory genes, inducible *NOS*, *IL-1β*, and *IL-6*, thereby mediating microglial polarization to the M2 (anti-inflammatory) phenotype and decreasing the retinal neovascular area in mouse OIR [393]. MiR-410 eye drops have been reported to treat pathological angiogenesis, probably by blocking VEGF-A expression, in another mouse model of OIR [394].

Several studies have uncovered the roles of some lncRNAs in ROP. Upregulation of lncRNA *MALAT1* expression was observed in OIR mice. Inhibition of *MALAT1* alleviates retinal neovascularization by suppressing the CCN1/Akt/VEGF pathway and cytokines, such as IL-1β, IL-6, and TNF-α, acting as a miR-124-3p sponge, and regulating early growth response protein 1 [395,396]. The inhibition of another lncRNA, *MIAT*, with small interfering RNA in an OIR mice model diminished retinal angiogenesis by downregulating the VEGF/PI3K/Akt pathway [397]. Finally, Wang et al. [398] observed the upregulation of lncRNA *TUG1* in mouse OIR. *TUG1* promoted retinal angiogenesis by sponging miR-299-3p and inducing VEGF-A. 

The roles of circRNAs in ROP have been reported in several studies. Expression of circZNF609, the first circRNA identified in mouse OIR, increases under hypoxia [399]. Inhibition of circZNF609 suppresses both retinal angiogenesis and vessel loss in vivo. The study highlighted the role of the circZNF609/miR-615-5p/myocyte-specific enhancer factor 2A axis in regulating endothelial cell function [399]. Deng et al. [400] observed a decrease in circPDE4B expression in OIR mice. CircPDE4B displays anti-angiogenic properties by inhibiting HIF1α and VEGF-A expression, sponging miR-181c, and modulating von Hippel–Lindau. Zhou et al. [401] analyzed the circRNA profiles of retinal samples from OIR and normal mice. Gene ontology analysis revealed that angiogenesis was the most upregulated biological process. Several circRNAs have been predicted to compete with endogenous RNAs using bioinformatics. Various circRNA–miRNA–mRNA regulatory axes have been implicated in ROP pathogenesis through reverse transcription-quantitative PCR validation.

Several miRNA-targeting drugs have been studied in clinical trials for the treatment of diseases, such as solid cancer, hepatitis C virus infection, and heart failure [402,403,404,405]. However, lncRNA- or circRNA-targeting therapeutics have not yet undergone any clinical trials. NcRNA modulation as a novel treatment for ROP may possess advantages, such as increased specificity for pro-angiogenic molecular pathways and less destruction. Nonetheless, several obstacles, including methods to enhance the stability of RNA therapeutics and deliver them specifically to target cells, remain to be overcome. Special chemical modifications may be applied, and encapsulated carriers, such as liposomes, may be adopted for this purpose [406,407].

#### 5.3.6. Gene Therapy

Evidence of the genetic contribution to the pathogenesis of ROP has been implied in studies of different OIR rat strains [408,409], human monozygotic and dizygotic twins [410,411], and preterm patients with extreme phenotypes [412]. Estimated heritability for susceptibility to ROP has been reported to be approximately 70–73% [410,411]. Several candidate genes modulating downstream inflammatory and angiogenic pathways leading to ROP have been identified [413] and may potentially become targets of novel therapeutics.

Commonly used viral vectors, such as lentiviruses, adenoviruses, and adeno-associated viruses, can deliver genes to the retina [414]. In mouse OIR, the transfer of anti-angiogenic genes, including pigment epithelium-derived factor, tissue inhibitor of metalloproteinase-3, and endostatin, via subretinal injection of adeno-associated viral vectors successfully alleviates retinal neovascularization [415]. Clinical trials of numerous viral vector-based gene therapies for managing proliferative retinopathy are currently underway. For instance, recombinant adeno-associated viral vectors encoding sFLT-1 were recently tested in a phase I clinical trial including patients with exudative AMD [416].

Another potential treatment may be genome engineering via clustered regularly interspaced short palindromic repeats (CRISPR)-CRISPR-associated protein 9 (Cas9) endonucleases. It may permanently suppress retinal angiogenesis by editing genes at the DNA level [417]. For example, CRISPR/CAS9-based depletion of VEGF-A, VEGFR2, and HIF-1α effectively blocks retinal or choroidal neovascularization in vivo and in vitro [418,419,420,421,422,423]. Given that growth factors are essential for ocular development throughout childhood, concerns may arise regarding the long-lasting effects of gene therapy. However, further research and clinical trials are required to determine the timing and optimal dosage of gene therapy to prevent any unfavorable toxicity and immunogenicity.

## 6. Conclusions

ROP is a complicated vasoproliferative vitreoretinal disorder that involves various angiogenic and inflammatory mediators. The two-phase disease model emphasizes the importance of assessing cytokine levels in a time-dependent manner in the evaluation of ROP pathogenesis. A specific cytokine may exert both pro- and anti-inflammatory and pro- and anti-angiogenic effects, depending on its dosage, time of administration, duration of treatment, and target tissue. Although BRB breakdown occurs in ROP pathology, the expression levels of cytokines often differ between the blood and the vitreous. Although various treatment modalities, such as cryotherapy, laser photocoagulation, and the use of anti-VEGF agents, have been well-established, novel therapeutics are required to curb the increasing prevalence of ROP. Understanding the cytokines at play and their association with other perinatal diseases may provide important insights to target the underlying pathways of ROP more precisely. Recently, suppression of disordered retinal angiogenesis via supplementation of EPO and its derivatives, PUFAs, and IGF-1/IGFBP-3 complex, modulation of HIF, and inhibition of Scg3 has attracted the attention of researchers for the treatment of ROP. Moreover, gut microbiota modulation, ncRNAs, and gene therapies have been shown to be potentially effective in regulating ROP. These emerging therapeutics may treat future preterm infants with ROP. 

## Figures and Tables

**Figure 1 jpm-13-00291-f001:**
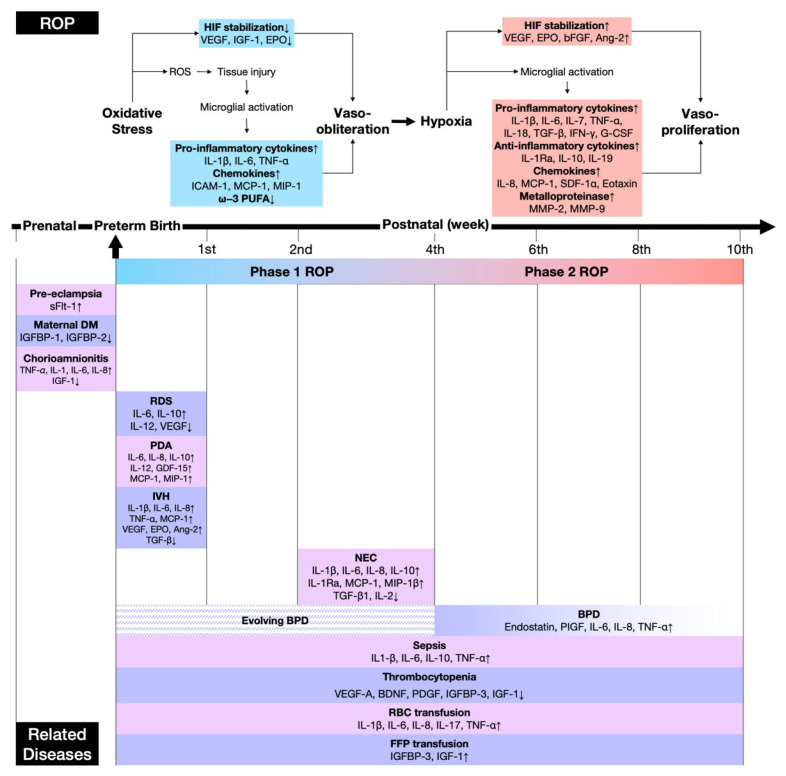
Common inflammatory and angiogenic mediators involved in the pathogenesis of retinopathy of prematurity (ROP) and the associated maternal and neonatal conditions. During phase I, oxidative stress suppresses pro-angiogenic factors. Hyperoxia-induced damage triggers over-activated inflammation, resulting in vaso-obliteration. During phase II, the hypoxic environment upregulates pro-angiogenic factors by hypoxia-inducible factor (HIF) and promotes both pro-inflammatory and anti-inflammatory cytokines. Anti-inflammatory cytokines may also display some pro-inflammatory properties. The subsequently upregulated metalloproteinase degrades the extracellular matrix, stimulates endothelial cell migration, and further recruits more growth factors, eventually causing vasoproliferation extending from the retina into the vitreous. The common onset timing of several maternal and neonatal conditions associated with the alternation of systemic cytokines is described below. These factors may contribute to ROP pathogenesis. Sepsis, thrombocytopenia, and blood transfusions are illustrated throughout the timeline since they can occur at any time. ↑: upregulation; ↓: downregulation. Abbreviations: ROP: retinopathy of prematurity; ROS: reactive oxygen species; VEGF: vascular endothelial growth factor; IGF-1: insulin-like growth factor-1; IGFBP: insulin-like growth factor-binding protein; EPO: erythropoietin; IL: Interleukin; TGF-ß: transforming growth factor-ß; TNF-α: tumor necrosis factor-α; ICAM-1: intercellular adhesion molecule-1; MCP-1: monocyte chemotactic protein-1; MIP-1: macrophage inflammatory protein-1; ω-3 PUFA: ω-3 polyunsaturated fatty acids; HIF: hypoxia-inducible factor; bFGF: basic fibroblast growth factor; Ang-2: angiopoietin-2; IFN-γ: interferon-γ; G-CSF: granulocyte colony-stimulating factor; IL-1Ra: interleukin-1 receptor antagonist; SDF-1α: stromal cell-derived factor-1α; MMP: matrix metalloproteinase; sFlt-1: soluble fms-like tyrosine kinase-1; PIGF: placental growth factor; BDNF: brain-derived neurotrophic factor; PDGF: platelet-derived growth factor.

**Table 1 jpm-13-00291-t001:** Systemic cytokines associated with Retinopathy of Prematurity.

Cytokines	Study Subjects	Expression in OIR/ROP Samples	Characteristics and Actions	Ref.
Angiopoietin-2	Mice Human	↑ (retina, vitreous)	Competes against angiopoietin-1, dephosphorylates Tie2 receptor, and initiates neovascularization Works in concert with VEGF	[141,146,147,148]
Endocan	Mice Human	↑ (retina, blood)	Promotes endothelial cell tube formation and vessel sprouting	[87,165]
Endostatin	MiceHuman	↑ (amniotic fluid)	Inhibits endothelial cell proliferation, migration, or invasion Inhibits capillary tube formation Inhibits retinal VEGF secretion	[128,161]
Eotaxin	Mice Human	↓ then ↑ (retina) ↑ (vitreous) ↓ (blood)	Attracts eosinophils and mediates inflammation Promotes angiogenesis via the receptor CCR3	[23,198,199]
EPO	Mice Human	↓ then ↑ (retina) ↑ (vitreous) ↓ or ↑ (blood)	Regulates hematopoiesis Displays neuroprotective, anti-apoptotic, and anti-oxidative properties Modulated by HIF-2 Signals through EPO receptor to promote angiogenesis	[72,73,74,84,103,104,107,108,109,110]
FGF	Mice Rats Human	↑ (vitreous, blood)	Elicits angiogenic effect via both VEGF-dependent and VEGF-independent pathways Activates STAT3 signaling pathway Potentially binds to VEGFR2 Displays neuroprotective effect in photoreceptor cells	[23,36,39,140]
G-CSF	Mice Human	↑ (retina, vitreous)	Regulates hemopoiesis and immunity Reduces oxygen-induced retinal vascular obliteration, and triggers angiogenesis in ischemic tissue Displays neuroprotective properties Takes part in synthesis of IGF-1	[23,49,211,212]
IFN-γ	Mice Human	↑ (aqueous)	Activates macrophages Exhibits proinflammatory, anti-viral, anti-proliferative, pro-apoptotic, and antitumor properties Impedes the proliferation, migration, and tube formation of endothelial cells via STAT1 signaling, even under VEGFA-treated condition	[23,58,203,204]
IGF-1	Mice Human	↓ then ↑ (retina) ↓ ( blood)	Promotes VEGF synthesis and maximizes its pro-angiogenic effect Promotes endothelial cell proliferation and integrity maintenance	[28,72,87,116,117,118,119,120,121,124]
IGFBP-3	Mice Human	↓ then ↑ (retina) ↓ (blood)	Serves as regulators and serum carriers of IGFs Preserves retinal vessels under oxygen-induced damage Facilitates vessel regrowth Decreases retinal neovascularization tufts formation	[29,31,117,126,127,128]
IL-1β	Rats Human	↑ (retina, choroid)	Sustains activation of retinal microglia Induces microvascular injury through release of semaphorin-3A from adjacent neurons Induces BRB breakdown Induces retinal ganglion cell death Induces choroidal involution and outer neuroretinal anomalies	[18,20,21,22,23]
IL-1Ra	Human	↑ (vitreous, tears)	Prevents IL-18 angiogenic effects Prevents IL-1β-induced cell death	[49]
IL-6	Human	↑ (blood)	Induces acute-phase protein response Displays both pro-inflammatory and anti-inflammatory properties	[27,28,29,30,31]
IL-7	Human	↑ (vitreous, blood)	Regulates the development and homeostasis of T cells, B cells, and natural killer cells Induces RPE-derived MCP-1 and IL-8	[23,36]
IL-8	Rats Human	↑ (retina, blood)	Promotes directional migration of neutrophils, basophils, and T lymphocytes Stimulates endothelial proliferation and capillary tube formation Inhibits endothelial cell apoptosis Enhances MMP-2, 9 and gelatinase activity	[30,38,39,40]
IL-10	Mice	↑ (retina)	Inhibits the release of TNF-α, MIP-1α, and RANTES in microglia Guides macrophage behavior to a pro-angiogenic phenotype under hypoxia	[42]
IL-17	Mice Human	↑ (retina) ↓ (blood)	Induction of immune response against bacteria and fungus	[27,45]
IL-18	Mice Human	↓ then ↑ (blood)	Displays immunoregulatory activity with both angiogenic and angiostatic properties Regresses retinal pathological neovascularization, rather than inhibiting its development	[27,46]
IL-19	Mice	↑ (retina)	Induces proliferation and migration of human retinal endothelial cells Promotes pro-angiogenic M2 macrophage polarization	[48]
MCP-1	Human	↑ (blood)	Induces migration and aggregation of monocytes/microglia Displays pro-angiogenic properties Recruits pro-angiogenic macrophages	[23,36,192]
MIP-1	Mice Human	↑ (retina, blood)	Induces migration and aggregation of monocytes/microglia Promotes revascularization of the avascular retinal area	[36,193,194]
Neurotrophins (BDNF, NT-4)	Mice Human	↓ (retina, blood)	Links the nervous and immune system (positive feedback autocrine loops) Promotes retinal ganglion cell survival after injury	[27,85,170]
PDGF	Rats Human	↓ (retina, blood)	PDGF-B helps recruit pericytes and vascular smooth muscles, stablizing vessels PDGF-CC intensifies MMP-2 and MMP-9 expression and augments monocyte migration	[85,151]
RANTES	Human	↑ (vitreous) ↓ (blood)	Contributes to innate immunity Recruits leukocytes to initiate inflammation	[23,27]
SDF-1α	Rats Human	↑ (retina, vitreous)	Triggers VEGF secretion by endothelial cells Promotes endothelial progenitor cell trafficking and cell migration	[83,181,182,183]
Soluble endoglin	Human	↑ (amniotic fluid)	Displays anti-angiogenic activity by downregulating TGF-β signaling Impairs retinal vascular growth	[128]
TGF-β	Rats Human	↑ (retina) ↓ (blood)	Displays both pro-angiogenesis and anti-angiogenesis properties depending on its concentration Regulates cell growth, differentiation, migration, and extracellular matrix production Up-regulates the expression of VEGFR-1	[27,133,135]
TNF-α	Mice Rats Human	↑ (retina, blood) ↓ (amniotic fluid)	Displays both pro-inflammatory and anti-inflammatory properties Induces BRB breakdown Induces retinal ganglion cell death	[20,21,28,30,53,54,57,58]
VEGF	Mice Rats Human	↑ (vitreous, aqueous) ↓ (tears) ↓ or ↑ (blood)	Activated by HIF-1α, and works in concert with other angiogenic mediators Regulates endothelial cells migration and survival Promotes abnormal vessel sprouting Increases vascular permeability Displays inner retinal neuroprotective properties	[3,13,40,58,61,69,70,72,73,74,82,83,84,85,86,87,88,89,90,91,92]

↑: upregulation; ↓: downregulation. Abbreviations: OIR: oxygen-induced retinopathy; ROP: retinopathy of prematurity; IL: interleukin; BRB: blood–retina barrier; RPE: retinal pigment epithelium; MCP-1: monocyte chemotactic protein-1; MMP: matrix metalloproteinases; TNF: tumor necrosis factor; MIP-1: macrophage inflammatory protein-1; VEGF: vascular endothelial growth factor; HIF: hypoxia-inducible factor; CCR: C-C chemokine receptor type 3; EPO: erythropoietin; IGF-1: insulin-like growth factor 1; IGFBP-3: insulin-like growth factor binding protein 3; TGF-β: transforming growth factor beta; VEGFR: vascular endothelial growth factor receptor; FGF: fibroblast growth factor; STAT3: signal transducer and activator of transcription 3; PDGF: platelet-derived growth factor; BDNF: brain-derived neurotrophic factor; NT-4: neurotrophin-4; RANTES: regulated upon activation, normal T cell expressed and secreted; IFN-γ: interferon-γ; G-CSF: granulocyte colony-stimulating factor.

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
