# Peer review of "Systemic Cytokines in Retinopathy of Prematurity"

_jpm, 2023, doi:10.3390/jpm13020291_

Round 1

Reviewer 1 Report

The authors give a broad overview of systemic cytokines involved in ROP and it's rodent models. The paper is well written and the article is citing recently published data.

In the Table 1 it would be good to mention in what type of samples (blood vitreous or retina) the expression was detected. 

In Table 1, some cytokines are marked for both human and mice/rat samples, but some only human, even it is shown in literature also in rodents. Is there a reason how the species are chosen? 

It is mentioned in the text that there are conflicting results between studies, and that depending the case, some cytokines can be either pro or anti-angiogenic. The authors should be careful when listing the cytokines either pro or anti-angiogenic in the table 1. 

The cytokines in Table 1 could be in alphabetical order. 

IL-17 is marked as downregulated in human ROP in the table 1 and in the text. In the figure 1 its expression has been marked as upregulated.

Row781: blinds --> binds

Reviewer 2 Report

Dear authors,

The article is well organized with clear and objective writing. The contents are well described with relevant information for the scientific community. I really enjoyed the article and have nothing more to add.
